# MVA Vectored Vaccines Encoding Rift Valley Fever Virus Glycoproteins Protect Mice against Lethal Challenge in the Absence of Neutralizing Antibody Responses

**DOI:** 10.3390/vaccines8010082

**Published:** 2020-02-12

**Authors:** Elena López-Gil, Sandra Moreno, Javier Ortego, Belén Borrego, Gema Lorenzo, Alejandro Brun

**Affiliations:** Animal Health Research Centre (CISA), National Institute for Agriculture and Food Research and Technology (INIA), Valdeolmos, 28130 Madrid, Spain; melenalopezgil@gmail.com (E.L.-G.); sandramorenofdez@gmail.com (S.M.); ortego@inia.es (J.O.); borrego@inia.es (B.B.); lorenzo.gema@inia.es (G.L.)

**Keywords:** Rift Valley fever virus (RVFV), modified vaccinia Ankara (MVA), cellular response, neutralizing antibodies, Gn Gc glycoproteins, passive serum:virus transfer

## Abstract

In vitro neutralizing antibodies have been often correlated with protection against Rift Valley fever virus (RVFV) infection. We have reported previously that a single inoculation of sucrose-purified modified vaccinia Ankara (MVA) encoding RVFV glycoproteins (rMVAGnGc) was sufficient to induce a protective immune response in mice after a lethal RVFV challenge. Protection was related to the presence of glycoprotein specific CD8+ cells, with a low-level detection of in vitro neutralizing antibodies. In this work we extended those observations aimed to explore the role of humoral responses after MVA vaccination and to study the contribution of each glycoprotein antigen to the protective efficacy. Thus, we tested the efficacy and immune responses in BALB/c mice of recombinant MVA viruses expressing either glycoprotein Gn (rMVAGn) or Gc (rMVAGc). In the absence of serum neutralizing antibodies, our data strongly suggest that protection of vaccinated mice upon the RVFV challenge can be achieved by the activation of cellular responses mainly directed against Gc epitopes. The involvement of cellular immunity was stressed by the fact that protection of mice was strain dependent. Furthermore, our data suggest that the rMVA based single dose vaccination elicits suboptimal humoral immune responses against Gn antigen since disease in mice was exacerbated upon virus challenge in the presence of rMVAGnGc or rMVAGn immune serum. Thus, Gc-specific cellular immunity could be an important component in the protection after the challenge observed in BALB/c mice, contributing to the elimination of infected cells reducing morbidity and mortality and counteracting the deleterious effect of a subneutralizing antibody immune response.

## 1. Introduction

Rift Valley fever virus (RVFV), a mosquito-borne bunyavirus, is widely distributed in Sub-Saharan countries, Egypt and the Arabian Peninsula, causing disease in both humans and livestock [1]. RVFV is considered an emerging threat for non-endemic countries due to the movement of infected animals and/or translocation of infected mosquitoes [2,3]. The ample range of RVFV competent mosquito vectors present in many areas of the Mediterranean basin suggests that RVF outbreaks in non-endemic areas could potentially end-up in the establishment of enzootic infection cycles [4]. Should this happen it would cause serious concern for both public and animal health. It is therefore desirable to develop control tools as well as enhance our knowledge about the immune mechanisms that correlate with the protection elicited by RVFV vaccines. The modified vaccinia Ankara (MVA) virus has been widely used as a carrier of vaccine antigens due to its safety and immunogenicity profile [5]. The MVA vector is a highly attenuated version of a vaccinia virus strain that has lost around 30 kb of sequence upon passage in primary avian cells (CEF) so that many host range and immune-modulatory genes are absent or nonfunctional [6]. This restricts the replication of the virus in most mammalian cells. Besides, it has been demonstrated that MVA itself is very immunogenic (induces humoral responses and is an excellent inducer of T-cell responses). It has been used in human preclinical and clinical trials [7] and more recently it has also been evaluated for different animal diseases, including zoonotic diseases [8]. In the case of zoonotic diseases the data obtained in animal field trials may also help to accelerate the development of corresponding human vaccines.

In order to control RVF it is generally accepted that the induction of neutralizing antibodies is an important correlate of protection [9]. Therefore, the Gn and Gc glycoproteins are the main vaccine antigen targets since both glycoproteins are membrane proteins forming spikes on the surface of the virions that can be accessible to neutralizing antibodies [10,11,12], precluding either internalization and/or nucleocapsid uncoating. Both glycoproteins are synthesized as a polyprotein precursor that becomes localized in Golgi membranes where they are glycosylated and cotranslationally processed by a yet unidentified cellular protease that releases one end of the membrane attachment sites, allowing both ectodomains to interact and the glycoproteins to acquire their final conformation on the surface of the virion [13,14]. The structure of GnGc architecture has been recently elucidated indicating that Gn shields Gc in the viral particles avoiding to expose the Gc fusion loop to antibodies [15] and offers an hypothesis for the neutralizing mechanisms of protective antibodies [16]. Our previous works using recombinant MVA (rMVA) as vector vaccines expressing both glycoprotein antigens (rMVAGnGc) showed an intriguingly low level of in vitro neutralizing antibody induction upon single dose administration, particularly when compared to other similar rMVA vaccines encoding RNA virus glycoprotein antigens [17]. In our system, the lack of a humoral protective antiglycoprotein response was observed in both mouse models and disease natural hosts [18]. Particularly, for sheep no neutralizing antibody induction was demonstrated either after one or two serial rMVAGnGc vaccine doses, questioning the efficacy of this vaccine in this species [18]. The lack of neutralizing antibody induction could be related to the vector platform used in those experiments (MVA) since the same coding glycoproteins expressed by means of an adenovirus vector induced a potent set of neutralizing antibodies [19,20].

In this work we show that rMVA vaccines expressing independently versions of the glycoprotein Gc or Gn were able to confer substantial protection in mice, albeit inducing no detectable in vitro neutralizing antibody responses. Rather, the protection observed upon challenge was related to a strong T-cell response, which appeared more prominent against Gc epitopes. Intriguingly, the immune response elicited by the rMVA vaccine encoding both glycoproteins or glycoprotein Gn endowed the serum with the capacity to exacerbate disease when a serum:virus mixture was passively transferred to naive mice, as shown in experiments using either mouse or sheep immune sera obtained from rMVA vaccinated animals. It therefore appears that evoking a strong cellular response counteracts the failure in inducing protective humoral responses when an MVA vaccine is used. Conversely, the absence of an effective cell-mediated immune response may lead to disease exacerbation when subneutralizing antibody responses are elicited. These results warrant the optimization of our rMVA vaccines towards the induction of optimal humoral responses.

## 2. Materials and Methods

### 2.1. Generation of Recombinant rMVA Encoding RVFV Gn and Gc Glycoproteins

The rMVA-GnGc recombinant virus generated by homologous recombination of wild type MVA DNA and a plasmid construct encoding RVFV-MP12 glycoproteins (GB accession: DQ380208.1) into the TK locus of MVA was described previously [17]. For the generation of rMVA-Gn and rMVA-Gc, the recombination (shuttle) plasmid encoding an RVFV-MP12 GnGc tagged sequence (plasmid #1389 produced at the Viral Vector Core Facility (VVCF), Jenner Institute, Oxford) was used as a template of an inverse PCR reaction using specific 5′ phosphorylated primers (Appendix A). After religation of the PCR fragments, two new shuttle vectors were generated, in which the Gn ectodomain (encoding amino acids Met131 to His580 of the translated polyprotein precursor) and a Gc including the C-terminal transmembrane-cytosolic tail (amino acids Cys690 to Ser1197) were placed under the control of the vaccinia p7.5 early/late promoter. The N-terminus of each recombinant polypeptide contained an in-frame fusion of the human tissue plasminogen activator leader sequence (tPA), known to enhance transgene expression and immunogenicity [21]. The C-terminus of each protein contained an H-2K^d^ restricted CD8+ T cell epitope from *Plasmodium berghei* circumsporozoite protein (pb9) and an antiV5 monoclonal antibody recognition sequence. The plasmid for MVA construction also includes GFP as a reporter gene under the control of the vaccinia p11 late promoter. Both shuttle vectors were transfected into DF-1 cells (ATCC-CRL-12203) using lipofectamine 2000 (Thermo Fisher Scientific, Waltham, MA, USA), then infected with parental MVA and homologous recombination allowed the insertion of either Gn ectodomain (eGn) or Gc ORFs and the GFP marker gene at the TK locus of the MVA. Three consecutive rounds of green plaque purification were performed in order to obtain a pure preparation of each recombinant virus. The recombinant viruses (named rMVAGn and rMVAGc) were then further expanded in DF-1 cells. Semipurified, concentrated, virus preparations were obtained upon ultracentrifugation of infected cell extracts in a 36% sucrose cushion. The sucrose-purified virus fractions were titrated into DF-1 cells and stored at −80 °C until use.

### 2.2. Western Blot Analysis

Expression of recombinant RVFV glycoproteins was analyzed by western blots of infected cell lysates using either specific antiGn or Gc antibodies [22] or a monoclonal antibody against V5 peptide tag (Bio-Rad, Hercules, CA, USA)). BHK-21 cells (ATCC CCL-10) were infected with the different recombinant MVA viruses described above, at 5 pfu/cell or were mock infected. At 24 h post infection the cells were harvested, pelleted, washed in PBS-containing protease inhibitor cocktail (Sigma-Aldrich, San Luis, MO, USA), and lysed with cytoplasmic extraction buffer (10 mM HEPES pH 7.9, 10 mM KCl, 0.1 mM EDTA, and 0.3% NP−40). After a centrifugation step to release intact nuclei, extracts were mixed with an equal amount of 2X Laemmli buffer, including DTT as a reducing agent and proteins were resolved in 12% SDS-PAGE and blotted onto nitrocellulose membranes. After a blocking step with 5% low fat dry milk in PBS (blocking buffer), antiRVFV Gn monoclonal antibody 84a (1:3000 dilution), monoclonal antiV5 tag (1:5000), or a rabbit antiGc polyclonal antibody (1:5000) were applied to membranes in blocking buffer with 0.01% Tween-20 and incubated for 1 h at room temperature. Horseradish peroxidase conjugated antimouse or antirabbit antibodies (1:5000) were incubated to the membranes after three washing steps with PBS Tween-20 (PBST). The resulting immunocomplexes were detected by enhanced chemiluminescence (GE Healthcare, Little Chalfont, Buckinghamshire, UK) and X-ray film exposure.

### 2.3. Indirect Immunofluorescence and Laser Confocal Microscopy

Cells were grown in either multi-well 96 (MW96) plates or in glass coverslips (CS) and infected with the recombinant MVA viruses at a multiplicity of infection (MOI) of 1. 24 h after infection the cells were fixed and permeabilized with 100% ice-cold methanol (MW96) or fixed with 4% paraformaldehyde and permeabilized with 0.5% Triton-X100 (CS). Fixed cells were blocked with 10% FBS in PBS (10% blocking solution) for 30 min at room temperature (rt). AntiV5tag mAb, glycoprotein specific antibodies or antibodies specific to ER and Golgi proteins calreticulin and human mannosidase II (Bio-Rad’s AHP516 and AHP674 antibodies) were incubated for 1 h at rt in 2% blocking solution with 0.01 Tween-20. After three serial washing steps with PBST Alexa 488 conjugated antimouse, or Alexa-Fluor 594-conjugated antirabbit or antigoat mabs (Thermo) were incubated for 30 min at rt. Stained cells in MW96 were visualized using a Zeiss AX10 inverted fluorescence microscope (Zeiss Gmbh, Oberkochen, Germany). Stained CS preparations were mounted onto glass microscopy slides with or without DAPI staining (Thermo Fisher) and were visualized analyzed in a Zeiss LSM880 confocal laser microscope. Images were further processed using the Zen Zeiss software.

### 2.4. Immunoprecipitation Analysis

For immunoprecipitation, monolayers of Vero cells were infected with RVFV-MP12 virus in the presence of 300 µCi/mL of [S35]-Methionine-Cysteine solution (Grupo Taper S.L, Alcobendas, Madrid, Spain). After 24 h the cells were lysed in radioimmunoprecipitation assay buffer (150 mM NaCl, 1% NP-40, 0.1% SDS, and 50 mM Tris, pH 7.5) and 20 µl of pooled sera from mice vaccinated with different rMVA viruses were incubated for 1 h at room temperature in a rotary shaker. Paramagnetic protein-G beads (Thermo Fisher) were added and incubated for an additional hour. The immunocomplexes were washed three times with the RIPA buffer and then separated by 12% SDS-PAGE. Fixed gels were subjected to fluorographic enhancement using Amplify solution (GE). After drying the gels were exposed to X-ray film.

### 2.5. Immunization, Sampling for Immunological Assays and RVFV Challenge

Groups of 5–10 BALB/c mice eight to ten weeks-old (Envigo RMS, Barcelona, Spain) were immunized intraperitoneally with 10^7^ pfu of sucrose-cushion purified rMVA in phosphate-buffered saline (PBS). One or two weeks postvaccination, blood samples were taken either for neutralization assays (serum) or IFN-γ ELISpot (PBLs), and splenocytes for ex-vivo IFNγ ELISPOT at 7 or 14 dpi (*n* = 4). The remaining mice (*n* = 5), together with additional groups of unvaccinated BALB/c mice and immunized with nonrecombinant MVA (MVA control, expressing only GFP), were all challenged intraperitoneally with 10^3^ plaque-forming units (pfu) of the South African RVF virus strain 56/74 [23]. The immunization and challenge studies were also performed in a similar manner using 129SvEv mice (B&K Universal Group Ltd, Hull, UK). To monitor viremia, blood samples were taken at 72 h after RVFV infection, and tested for virus isolation on cell culture as described [24]. Briefly, blood dilutions were incubated with Vero cells cultures and examined for the viral cytophatic effect (cpe). After 96 h the extent of cytophatic effect was recorded and cells were fixed and stained with 2% Crystal Violet in 10% formaldehyde solution. The extent of cpe was quantified to estimate a tissue culture infective titer (TCID_50_). Serum samples collected at later times postinfection in surviving mice were analyzed for the presence of neutralizing antibodies as described below. Vaccine efficacy estimation was evaluated in terms of morbidity and mortality monitoring daily over three weeks. All surviving mice were culled after 21 days of follow-up. Procedures involving animals received institutional approval (INIA’s ethics and biosafety Committee) as well as granted permits from regional veterinary authorities (Comunidad de Madrid PROEX 108/15).

### 2.6. Assessment of RVFV Serum Neutralizing Antibodies

Serum neutralizing antibody titers were measured in Vero cell monolayers by serial dilutions of serum mixed with an equal volume of medium containing MP-12 RVF virus strain and incubated for 1 h at 37 °C. After 72–96 h the cells were fixed and stained in a solution containing 10% formaldehyde and 2% crystal violet in PBS. The neutralization titer defined as the highest serum dilution at which cell lysis was reduced by 50% relative to cells incubated with RVF virus only. The assays were performed in triplicate and scored by an operator blinded to the vaccination regimen.

### 2.7. Analysis of T-Cell Responses Against RVFV Glycoproteins

Viral glycoprotein-specific T cells were measured by ex vivo IFNγ ELISPOT assay on splenocytes and pooled PBLs as described [17]. Gn, Gc-specific and nonspecific class-I restricted peptides were used for restimulation at a final concentration of 5 µg/mL in all assays for 18 h. IL-2, IL-6, IL-4, and IL-5 cytokine capture ELISAs (BD Pharmingen) were also performed using supernatants from peptide restimulated spleen cell cultures. Known concentrations of mouse IL-2, IL-6, IL-4, or IL-5 were used to generate a standard curve to correlate optical densities with cytokine concentration. The sensitivity limit of the assay was estimated in (61.25 pg/mL for IL-6, 250 pg/mL for IL-2, 3.75pg/mL for IL-4, and 37.5 pg/mL for IL-5).

### 2.8. Passive Transfer of Antibodies

The sera for passive transfer protection studies were generated by pooling sera from mice immunized with the different rMVA constructs expressing the same antigens. Serum pools were prepared from day 14 post immunization and analyzed by virus neutralization and immunoprecipitation assays. As a positive control, antiRVFV immune mouse serum was used while antihuman adenovirus 5 (AdHu5) pooled mouse serum (collected also at 14 days post immunization) was used as a negative control serum. For passive protection experiments each serum pool was ten-fold diluted in the virus inoculum used to challenge each group of 5 animals. 100 µL of each virus/serum mixture was injected intraperitoneally into adult female BALB/c mice. The virus challenge dose per mouse corresponded to 5 × 10^3^ pfu. Animals were monitored for clinical signs and mortality during three weeks and were weighed daily to quantify the extent of morbidity after challenge. Additionally, serum from sheep immunized with the rMVAGnGc vaccine, rMVA or from mock vaccinated was also pooled and passively transferred to mice.

### 2.9. Statistical Analysis

The log rank (Mantel–Cox) test was used to check for differences in survival analysis following RVFV challenge. Individual ELISPOT values were determined by subtracting background values obtained after stimulation with media only. Statistical significance was calculated by one-way analysis of variance (ANOVA) transforming ELISPOT counts to log_10_ to limit the range of variation found among individual mice. All analyses were done using the GraphPad 6.0 software (San Diego, CA). Differences were considered significant when *p* value <0.05

## 3. Results

### 3.1. Expression of Recombinant Gn and Gc Glycoproteins in rMVA Infected Cells

Expression was assessed by western blot analysis (Figure 1A). Both glycoprotein sequences were tagged with the V5 epitope sequence to compare their relative expression levels. It was observed that the infection of cells with both recombinant MVAGn and rMVAGc rendered detectable expression levels for both glycoproteins. This ruled out the possibility of low level expression conditioning the immunity conferred by each vaccine. Expression of glycoproteins was also confirmed using antiGn or Gc specific antibodies [22]. The detecting signal was similar to that of RVFV MP12 infected BHK-21 cells. The size of Gn expressed by rMVAGn was in accordance with its theoretical mass (50.6 kDa) but slightly lower when expressed by rMVAGnGc. Gc expression was also in good agreement with the expected size and similar in size to the one expressed by RVFV infection, although a smaller truncated polypeptide was also evident using the antiGc or the antiV5 tag antibodies. Detection of the expressed antigens was performed also by immunofluorescence assay (IFA)of rMVA infected Vero cell monolayers with an antiV5 tag monoclonal antibody. The subcellular staining pattern of each glycoprotein was in good agreement with intracellular membrane trafficking as it has been described for both glycoproteins (Figure 1B). Gn expression was predominantly cytoplasmatic with no evident association with endoplasmic reticulum (ER). In contrast Gc interaction with ER structures was more obvious as shown by the colocalization with the ER marker calreticulin. No clear association of Gn or Gc was found with Golgi structures at least at the time point assayed (24 hpi), as evidenced by the lack of costaining with an antihuman mannosidase-II mAb.

### 3.2. Efficacy Assessment of MVA Vaccines in Mice

The protective ability of a single dose of our rMVA vaccines was tested in BALB/c mice (Figure 2A). Mice immunized with rMVAGc virus showed an 80% survival after challenge. In this group at 11 dpi one mouse showed signs compatible to a delayed-onset neurological disease, dying at day 14 pi. In contrast, two of the mice vaccinated with rMVAGn showed earlier clinical disease dying at day 4 and 6 post challenge respectively. As expected, in the group of mice vaccinated with the rMVAGnGc construct the survival after challenge was 100%, with only one animal showing mild clinical signs between 3 and 4 dpi. In the mice from both control groups (either nonrecombinant MVA and unvaccinated) the mortality was 80% and 100% respectively with an earlier onset of disease in both groups. Differences in the survival rates observed for each group were statistically significant (χ^2^ = 09.503; df = 3; *p* = 0.023) when compared to the control MVA vaccine (Mantel–Cox log-rank test).

Since the protective effect of our rMVAGnGc single dose vaccine relies mainly in the induction of specific CD8+T-cell responses [17] we questioned whether its efficacy in a different inbred mouse strain would be compromised. Thus we tested the protective ability of our rMVA vaccines in the context of a different genetic background by using the 129SvEv mouse model (H-2^b^-haplotype). The sensitivity of this mouse strain to the RVFV challenge is higher than that of the H-2^d^ BALB/c strain (our unpublished data). Survival rates upon the RVFV challenge in the 129SvEv mice immunized with rMVAGnGc reached 80% with only one animal dying at 5 dpi. Contrarily, to what was observed in the BALB/c experiments, all of the 129SvEv mice that were vaccinated with rMVAGc or rMVAGn died upon challenge, with a slight delay in mortality in the rMVAGc group with respect to the MVA control group (Figure 2B). Differences in survival times of rMVAGnGc were highly significant (χ^2^ = 17.48; df = 3; *p* = 0.0006).

### 3.3. Analysis of Humoral Responses in rMVA Vaccinated Mice

We had previously reported the low level of in vitro neutralizing antibody induction induced by a rMVAGnGc vaccine in BALB/c mice and sheep [17,18]. As expected, levels of neutralizing antibodies in the serum from immunized BALB/c mice remained below the established detection threshold (1.3 log10, 1:20 serum dilution; Figure 3A).

Only two 129SvEv mice from the rMVAGnGc group showed titers slightly above the threshold limit (1.6 log10, serum dilution 1:40). The rest of mice, either vaccinated with rMVAGn or rMVAGc did not show titers above the detection limit in any of the mouse models used. Moreover, one animal from the MVA control group showed a VNT_50_ titer of 1:20 indicating that the observed neutralization at this dilution could be unspecific. When a more stringent neutralization determination was applied (i.e., VNT_100_), no single prechallenge serum showed neutralization in all microtiter wells (not shown). Upon RVFV 56/74 challenge, all surviving animals showed elevated neutralization titers reaching around 3 logs. Although these data could indicate a successful priming of the vaccines, a similar titer observed in a surviving mouse from the control group ruled out this possibility. Interestingly, the mean postchallenge neutralization titer of the rMVAGnGc vaccinated 129EvSv mice was slightly higher, in agreement with the two mice showing prechallenge neutralization titers over the sensitivity limit. In spite of the lack of a clear in vitro neutralization activity, both vaccines induced antibodies able to label RVFV-MP12 infected cells as shown by indirect immunofluorescence (Figure 3B). In the case of the rMVAGc serum, antiGc antibodies developed later, since a clearly positive fluorescent signal on infected cells was only detected in serum collected 14 days post immunization (Figure 3B). However, none of these prechallenge sera was able to inmunoprecipitate metabolically labeled RVFV glycoproteins in infected cell extracts (Figure 3C).

### 3.4. Analysis of Cellular Immune Responses to Vaccination

ELISPOT assay using 14 dpi pooled peripheral blood leukocytes (PBLs) from BALB/c mice immunized with rMVAGc showed the highest numbers of IFN-γ secreting cells upon restimulation with two different Gc-specific, MHC-I-restricted, peptides #13 (SYKPMIDQL) and #14 (GGPLKTILL; Figure 4A).

Accordingly, lower numbers of cells were found upon stimulation with the Gn specific peptide (SYAHHRTLL). The rest of the groups showed lower numbers of IFN-γ secreting cells, with the only exception of the rMVAGn group restimulated with the pb9 control peptide. The highest number of IFNγ secreting cells was also found in the MVAGc group in an ELISPOT assay using splenocytes collected at 7 dpi (Figure 4B) or 14 dpi (data not shown). In contrast to the PBL assay, the number of spots was higher for the rMVAGnGc and rMVAGn upon restimulation with a Gn specific peptide. Higher responses were observed upon stimulation with pb9 control peptide in the rMVAGn group in comparison with rMVAGc or rMVAGnGc groups. Of note, the rMVAGn PBLs and spleen cells were also stimulated with Gc peptide 14, although at lower levels. The peptides used for restimulation of BALB/c spleen cells were not able to stimulate IFNγ secretion in 129SvEv spleen cells (not shown), indicative of the restriction imposed by the specific haplotypes. The ELISPOT data correlated with the higher secretion of IL-2 and IL-6 cytokines, involved in T-cell survival that were detected by ELISA in the supernatants of restimulated cultures (Figure 5), indicative of the induction of a lymphoproliferative environment. Again, the group vaccinated with the rMVAGc vaccine displayed the highest amounts of both cytokines. On the other hand, IL-4 or IL-5, two of the main cytokines involved into B-cell proliferation, class switching, and differentiation to effectors were not detected in the same supernatants (not shown).

### 3.5. Assessment of Efficacy of Humoral Responses by Passive Serum Transfer Experiments

In order to gain insights in the role of the humoral response induced by the rMVA vaccines, a passive serum transfer experiment was designed. In this experiment individual sera from BALB/c mice collected 14 days after immunization with the different rMVA vaccines were pooled. The challenge virus dose (5 × 10^3^ pfu/mouse) was preincubated in the presence of each serum pool (final serum dilution 1/14) for 30 min, prior to the inoculation of mice. All mice that received the challenge dose in the presence of RVFV convalescent serum survived with no clinical display nor significant weigh loss (Figure 6A and Appendix A). In contrast, all mice from the rMVAGn group and most of the mice inoculated with serum pooled from the rMVAGnGc or Ad5 control vaccinated groups died shortly after inoculation (Figure 6A). Interestingly, four out of five mice transferred with virus plus donor serum from rMVAGc or MVA control survived longer than mice from the rMVAGnGc group (χ^2^ = 12.11; df = 3; *p* = 0.0070) and eventually recovered from infection (Appendix A). Accordingly with the survival data, mean viremia titers were more elevated in rMVAGn and rMVAGnGc groups when compared to rMVAGc or MVA control (Figure 6B), and the differences between means were statistically significant (*p* < 0.01, ANOVA test). Of note, the surviving mouse from the rMVAGnGc group had no conclusive viremia determination but it did not seroconverted (not shown), suggesting that this mouse was not efficiently infected.

In order to confirm these observations a second passive transfer experiment was carried out with a serum pool obtained from a different rMVAGnGc vaccination experiment. Again, the mortality rates were higher and occurred earlier in the rMVAGnGc group than in the control groups (not shown). These data were suggestive of an exacerbating disease effect induced by the serum from animals vaccinated with rMVAGnGc or rMVAGn vaccines. Intriguingly, the extended survival in the rMVA control group was totally unexpected, indicating that the protective effect in the mice was not related to the presence of antiRVFV specific antibodies. In order to check whether the enhanced pathogenic effect of the rMVAGnGc serum was not exclusive of the mouse immune serum, sheep serum pooled from a previous rMVAGnGc vaccination experiment [25] was also used in a similar transfer experiment. The results showed unequivocally an accelerated mortality, with statistical significance (χ^2^ = 7.740; df = 2; *p* = 0.0209) in the mice transferred with rMVAGnGc serum with respect to the serum from naive sheep (Figure 6C). Taken together the results observed suggest that the presence of serum anti Gn antibodies may trigger deleterious effects enhancing the infectivity of the virus inoculum.

## 4. Discussion

We proposed previously that the protective ability of a recombinant MVA vaccine encoding GnGc antigens relied mostly in T-cell immune responses in the absence of a strong in vitro neutralizing antibody response [17]. Apparently, the lack of neutralizing responses could be due to the type of immunity elicited by the vector itself, since the same coding sequence, either expressed by plasmid DNA, subunit vaccine (Gn) or delivered by means of an adenovirus vector eventually elicited stronger neutralizing antibody responses in mice [17,19,26]. In this previous work several MHC class-I restricted peptides from the glycoprotein sequences were identified for their ability to stimulate the secretion of IFNγ by CD8-T cells [17]. Here, our data confirms that BALB/c mice can be also protected upon the RVFV challenge by rMVAGc and, to a lesser extent, by rMVAGn and that this protection can also be achieved in the absence of neutralizing antibodies. According to the role of a cell-mediated immune response, the protection was restricted to a specific genetic background, as shown by the lack of survival upon challenge of 129SvEv mice immunized with the same vaccines (rMVAGn or rMVAGc). The detection of IL-2 and IL-6 supports the induction of cellular responses since both cytokines play a role in T-cell survival and activation. Particularly, Gc-specific T-cell responses may act as a key component in the protection after challenge observed in the rMVA immunized mice, perhaps contributing to the efficient elimination of RVFV infected cells. Our data also point out that the simultaneous expression of both glycoproteins by the MVA vector is an essential requirement for the induction of a protective response in the 129SvEv mouse strain. At this point it could be interesting to explore further how the genetic background determines the efficacy of the immune response and how differences in susceptibility to RVFV challenge may account for the observed differences in efficacy.

One of the most striking findings in this work is the exacerbating effect of some rMVA immune serum in infectivity. This was somewhat unexpected but may help to explain our previous observations in experiments conducted to evaluate the efficacy of the rMVAGnGc vaccine in sheep [25]. An indirect measure of viral replication in the host is the induction of antibody responses to immunogenic epitopes. For RVFV, the most immunogenic epitopes lie in the viral nucleoprotein N. Therefore, detection of antiN serum antibodies reveals the existence of a productive RVFV infection in the host. Earlier antiN antibody detection was observed in the serum from sheep vaccinated with rMVAGnGc when compared to non- and mock vaccinated controls. In addition, the amount of viral RNA detected in blood was higher at early times upon infection than in controls, indicating faster virus replication. We could reproduce here similar results upon the passive transfer of both mouse and sheep serum. One explanation to these findings would be the induction of subneutralizing antibodies able to enhance rather than block virus replication. Antibody dependent enhancement (ADE) has been described in several viral systems, with more detail in flaviviral infections [27,28,29,30,31,32]. In the case of RVFV it could be suggested that subneutralizing antiGn antibodies could bind to exposed Gn epitopes on the virus particle. In this scenario internalization of virus-antibody immune-complexes would be augmented in cells bearing complement or Fc receptors, increasing virus uptake and pathogenesis. This could explain the increased mortality of mice that were transferred with rMVAGnGc or rMVAGn serum:virus mixtures compared to that of rMVAGc or the MVA control. However, the protection observed in mice receiving the MVA control immune serum is puzzling. A plausible explanation could be that other nonantibody mediated humoral effectors provide some degree of protection although not sufficient to avoid deleterious effects of subneutralizing antibody responses. Thus, MVA or rMVAGc serum transfer would provide such antiviral effect while the transfer of rMVAGnGc or rMVAGn serum would enhance infectivity through subneutralizing antiGn antibodies. Both the classical and alternative pathways of the complement system can be activated upon viral infections and it has been shown that complement system plays an important role in poxvirus immunity [33]. On the other hand, deposition of complement proteins on the surface of enveloped virions enhances uptake by phagocytosis and potentially interferes with receptor interactions, virus entry, and uncoating [34]. Whether this hypothesis is or not true would deserve further experimentation. Nonetheless, it becomes clear that improving the quality of the antibody response of our MVA vaccines would render them more efficacious against a lethal RVFV challenge. Current work is underway to test the ability of novel MVA recombinants using different vector source and stronger promoter sequences for increasing antigen expression and enhancing proper processing of antigens.

## 5. Conclusions

In conclusion we confirmed the possibility of protecting mice against a lethal RVFV challenge without induction of neutralizing antibody responses, stressing the importance of cell-mediated immune responses in protection. Most importantly, failing in inducing proper neutralizing antibody responses may result in enhanced pathogenesis when the cell mediated immune response is impaired or absent.

## Figures and Tables

**Figure 1 vaccines-08-00082-f001:**
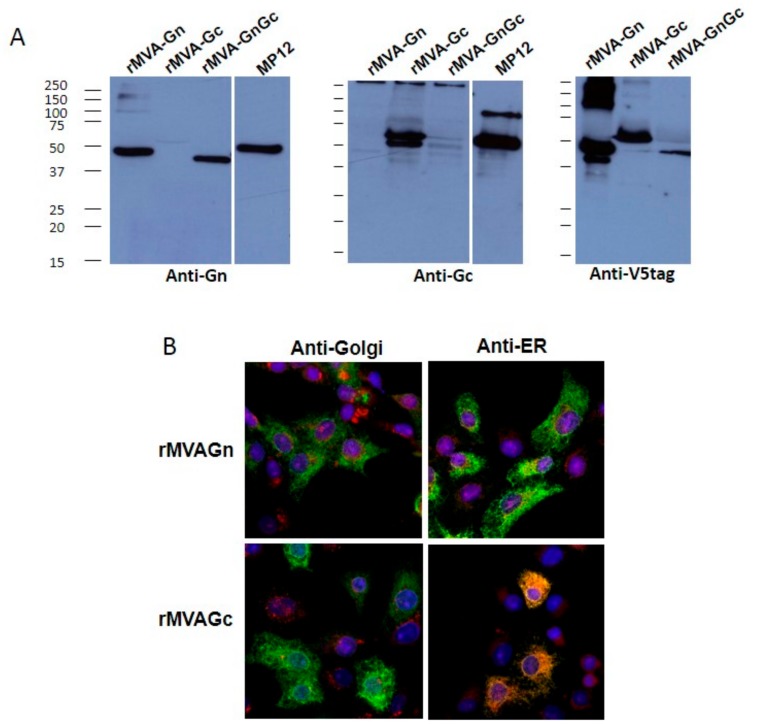
Expression and subcellular localization of recombinant Gn and Gc upon modified vaccinia Ankara (MVA) infection. (**A**). Western blot of different MVA infected BHK-21 cell extracts probed with mAb 84a antiGn or a rabbit polyclonal serum antiGc. The antiV5 tag mAb was used to compare the Gn and Gc expression levels and to confirm the expression of the full-length antigen. As a positive control a RVFV-MP12 infected cell extract was used. Numbers indicate relative molecular mass in kilodaltons. (**B**). Confocal immunofluorescence images of MVA infected Vero cells. Expression of Gn or Gc was detected with anti V5 tag mAb (green). Intracellular membranes were labeled with either antihuman mannosidase-II (Golgi) or anticalreticulin (ER) mAbs (red fluorescence) as indicated. Nuclei were labeled with DAPI stain (blue). All panels correspond to merged fluorescence images. Colocalization of Gc and ER membranes is evidenced by yellow-orange fluorescence.

**Figure 2 vaccines-08-00082-f002:**
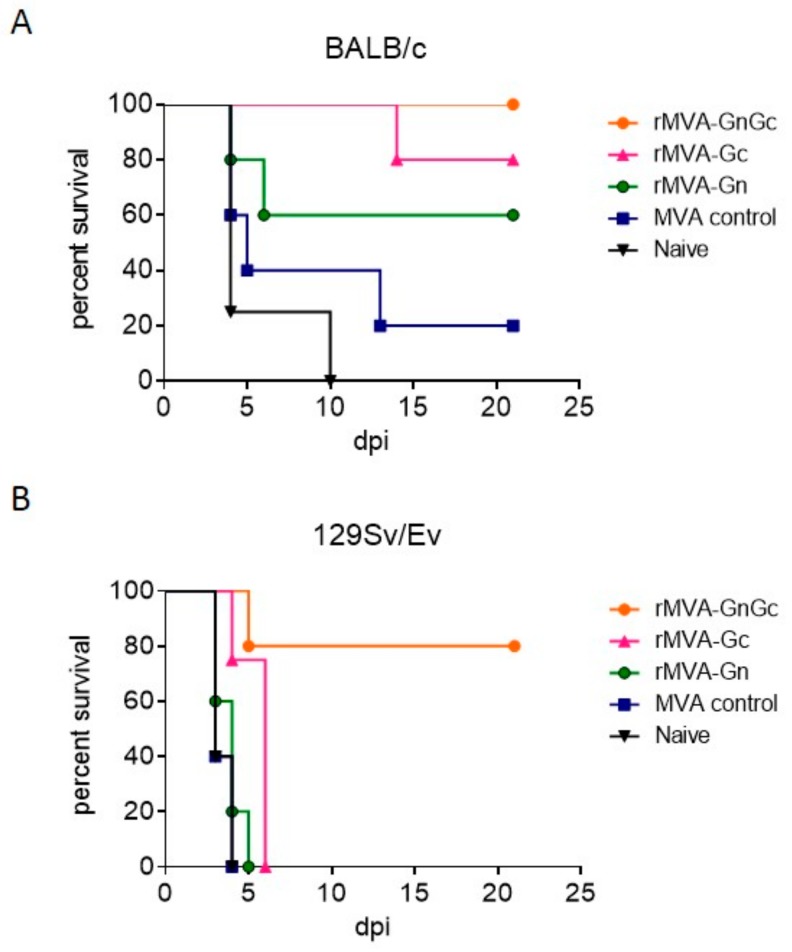
Survival of MVA vaccinated mice (*n* = 5) upon challenge with virulent Rift Valley fever virus (RVFV). Kaplan–Meier plots of BALB/c mice (haplotype H2^d^ (**A**) or 129EvSv mice (haplotype H2^b^) (**B**). The mice were vaccinated with a single intraperitoneal dose of 10^7^ pfu of each recombinant virus or were mock-vaccinated (naive). Two weeks after immunization the mice were challenged with 10^3^ pfu of RVFV 56/74. The mice were monitored for 3 weeks for the presence of signs of disease.

**Figure 3 vaccines-08-00082-f003:**
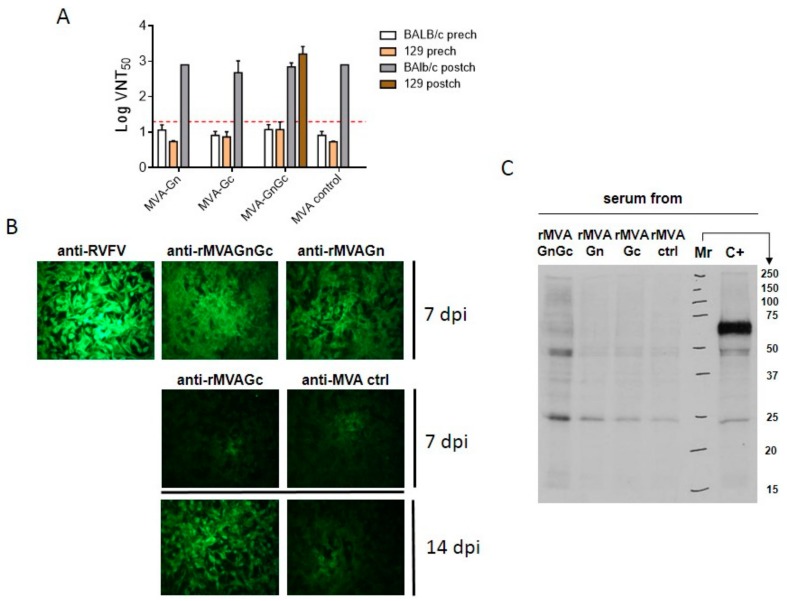
Humoral responses upon MVA vaccination in mice. (**A**). Virus neutralization titers (VNT) in serum samples taken 14 days after immunization (prech.) of either BALB/c or 129SvEv mice. Neutralization titers were also estimated in mice that survived the challenge (postch.). Bars represent mean plus SD. (**B**). Detection of RVFV-infected cells by IFA with serum from MVA vaccinated mice at 7 or 14 days post immunization (dpi). The figure shows representative images of viral plaques detected on cells. (**C**). Immunoprecipitation of RVFV-infected BHK21 cell extracts with serum from MVA vaccinated BALB/c mice. C+: positive control serum from mice immunized with an adenovirus vector encoding GnGc. Mr: relative mass in kDa.

**Figure 4 vaccines-08-00082-f004:**
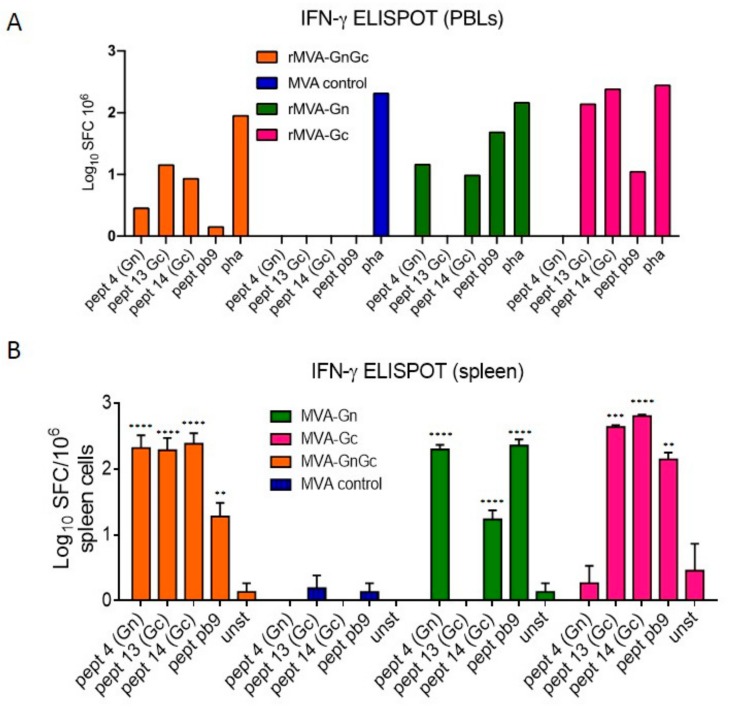
Cellular responses upon MVA immunization. (**A**). Interferon gamma ELISPOT assay of pooled (*n* = 5) BALB/c peripheral blood leukocytes (PBLs) taken at day 14 postimmunization with the different MVA vaccines. Each pool was restimulated with either Gn (#4), or Gc (#13 or #14) specific peptides or with peptide pb9. Nonspecific stimulation was induced with phytohemaglutinin (pha). (**B**). Mean ± SD log spot forming cells (SFC) values obtained in spleen cells from BALB/c mice (*n* = 2) at day 7 post MVA immunization. As above, the peptides 4, 13, and 14 were selected on the basis of their ability to stimulate Gn and Gc specific T-cell responses. Cell culture medium with no added peptide (unst) was used to measure the background of the assay. The pb9 peptide was used as a specific positive control for each recombinant MVA (rMVA) vaccinated mice. In all groups asterisks indicate significance levels for each peptide when compared to the unstimulated control (unst) using Dunnett’s multiple comparisons test (** *p* < 0.01; *** *p* < 0.001; **** *p* < 0.0001).

**Figure 5 vaccines-08-00082-f005:**
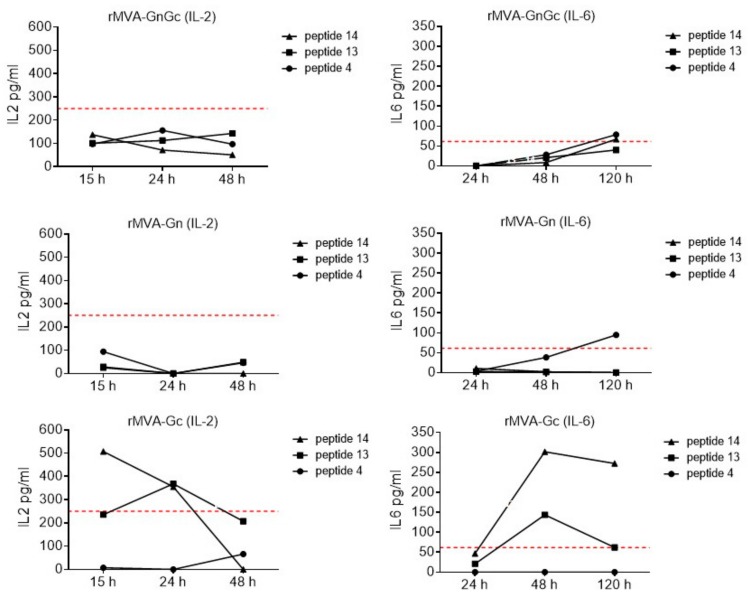
Detection of IL-6 and IL-2 secretion in ex-vivo restimulated spleen cells with peptides. The levels of each cytokine were estimated, using a capture ELISA, in supernatants collected at different times after stimulation. A standard curve was generated to correlate ELISA absorbance values with cytokine concentrations. The graph represents values after background subtraction from nonstimulated cells. The red dotted line determines the lower range of the standard curve.

**Figure 6 vaccines-08-00082-f006:**
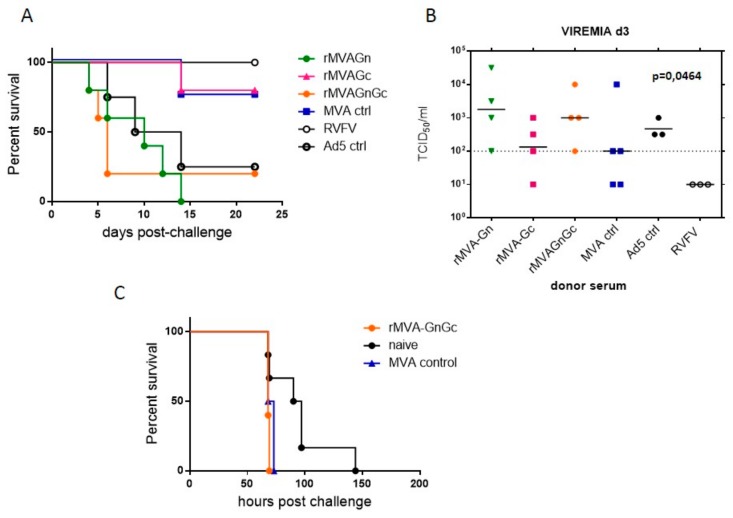
Effect of rMVA immune serum in BALB/c survival. Mice (*n* = 5) were passively transferred with a mixture of immune serum from MVA vaccinated mice and 5 × 10^3^ pfu of virulent 56/74 RVFV. For positive and negative control groups *n* = 4 was used. (**A**). Kaplan–Meier plots of survival proportions. The mice were monitored for 3 weeks for the presence of signs of disease. (**B**). Viremia at day three post inoculation tested by tissue culture infection doses in Vero cells. The infectious titer of each sample is defined as the reciprocal of the highest dilution of serum where a 50% of the cytophatic effect (cpe) is observed relative to noninfected controls. Only samples allowing a clear cpe determination are included. Samples inducing a non cpe-like effect were excluded. When no evident cpe was observed an arbitrary value of 10^1^ TCID_50_ below the sensitivity limit (10^2^) was assigned. Black lines represent means. Dotted line represents the sensitivity of the assay. The ANOVA test *p* value for differences among means is indicated. (**C**). Kaplan–Meier plots of BALB/c survival upon transfer with the rMVA ovine immune serum.

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
