# Peer review of "MVA Vectored Vaccines Encoding Rift Valley Fever Virus Glycoproteins Protect Mice against Lethal Challenge in the Absence of Neutralizing Antibody Responses"

_vaccines, 2020, doi:10.3390/vaccines8010082_

Round 1

Reviewer 1 Report

The authors evaluate different RVFV MVA vaccines in mice and determined the mechanism of protection is primarily cell mediated immunity for these vaccines.

Minor issues.

Line 337 Assessment of efficacy of humoral responses by passive serum transfer experiments

The passive transfer experiments demonstrate that the sera from the all the rMVA vaccines including the vector were not effective in protecting mice from RVFV whereas RVFV convalescent sera was effective in the protecting mice against RVFV infection.

Since the control rMVA which has no RVFV specific antibodies in it there is likely some other effect of the sera such as IFN which may be controlling the RVFV infection.

I do not see any biological significance between the groups as the only group that showed protection is the RVFV convalescent sera. 

Figure 6 B the viremia differences are not biologically significant between the different MVA vaccine sera groups with less than a log difference. 

Line 371 The results showed unequivovally an accelerated mortality, with statistical significance in the mice transferred with rMVAGnGc serum with respect to the serum from naïve sheep.

It does not seem that there is a huge biological difference in the time to death of 3 days to 4 days. Is one day faster really biologically significant??

Line 416 Other effector molecules IFNs the role of complement with RVFV

5 conclusions The different MVA vectored RVF vaccines in BALB/c mice all showed some protection and no enhancement of RVFV. This should be discussed. The role of primed B cells should be discussed as the conclusion is based on sera transfer in the absence of the B cells.

Author Response

RESPONSE TO REVIEWER 1 :

Line 337 Assessment of efficacy of humoral responses by passive serum transfer experiments

The passive transfer experiments demonstrate that the sera from the all the rMVA vaccines including the vector were not effective in protecting mice from RVFV whereas RVFV convalescent sera was effective in the protecting mice against RVFV infection.

Since the control rMVA which has no RVFV specific antibodies in it there is likely some other effect of the sera such as IFN which may be controlling the RVFV infection.

I do not see any biological significance between the groups as the only group that showed protection is the RVFV convalescent sera.

Figure 6 B the viremia differences are not biologically significant between the different MVA vaccine sera groups with less than a log difference.

The reviewer is right in that the RVFV convalescent serum transfer was more effective in protection (100% survival). However we think that an 80% percent protection from mortality provided by the MVA or MVAGc serum transfer can be considered a good efficiency. We agree in that other factors present in the sera may be responsible for the protection observed, something that at this moment we have not characterized, but that definitively deserves further investigation. It could be possible that antiviral molecules (including IFNs) present in the serum could trigger an antiviral state upon transfer in the mice sufficient to decrease the virus infectivity, but this needs to be empirically determined. With respect to the viremia we just limited to describe that the levels of infectivity found in blood are lower in the groups that showed higher survival rates. Therefore, it appears to be a correlation between the degree of survival and the presence of virus on blood samples. We understand that the number of animals is low to provide more power to our significance analysis but we still think that there is a clear trend towards a difference between the group means.

 Line 371 The results showed unequivocally an accelerated mortality, with statistical significance in the mice transferred with rMVAGnGc serum with respect to the serum from naïve sheep.

It does not seem that there is a huge biological difference in the time to death of 3 days to 4 days. Is one day faster really biologically significant??

We agree one day is not a huge difference but we would like to stress the fact that the disease signs manifested earlier in the mice (within hours of difference) with respect to the other groups. We mean that the progression of the disease is faster in the mice transferred with the serum.

 Line 416 Other effector molecules IFNs the role of complement with RVFV

Sorry. We do not understand well this sentence

5 conclusions The different MVA vectored RVF vaccines in BALB/c mice all showed some protection and no enhancement of RVFV. This should be discussed. The role of primed B cells should be discussed as the conclusion is based on sera transfer in the absence of the B cells.

This is an interest observation. Upon immunization we could expect some sort of enhancement of the infection but all MVA vaccines showed some degree of protection upon challenge. A putative explanation for not seeing such enhancement is that the vaccinated mice also evoked a protective cellular response that could counteract the potential harmful effect of the suboptimal humoral response. This is what we meant with the sentence “Most importantly, failing in inducing proper neutralizing antibody (i.e having a deleterious serum) responses may result in enhanced pathogenesis when cell mediated immune response is impaired or absent.”

Reviewer 2 Report

In the manuscript “MVA vectored vaccines encoding Rift Valley fever virus glycoproteins protect mice against lethal challenge in the absence of neutralizing antibody responses” the authors extend on their previous findings showing protection of mice from RVFV challenge using MVA-vectored RVFV GnGc vaccine. This vaccine induced potent T cell responses against RVFV GnGc while neutralizing antibody responses were minimal. In the current study the authors investigated the individual contribution and nature (T cell or humoral) of Gn or Gc specific responses in mediating protection against RVFV challenge.  The main finding of the study is that immune responses against Gc epitopes are more efficient at eliciting protection against RVFV and similarly to previous work with whole GnGc, are mediated by T cell and not neutralizing antibody responses. However, not all the conclusions of the study are justified by the data presented and significant modifications are required for this study to be publishable.

Major criticisms:

Fig. 4: What is the rationale of choosing the shown Gn and Gc peptides as stimulants. No explanation is given. Similarly, the pb9 positive control peptide is not described. Why are the T cell responses negative for pb9 peptide in MVA-control vaccinations?

Fig. 4A: What is the added value of showing PBL responses in the addition to splenocytes in 4B? Since there no replicates and no statistically significant differences can be determined, this data feels redundant.

Fig. 6. As presented, the conclusions from experiments shown in Fig.6 are not warranted. The major issue of this experiment is that mice, which have been passively transferred with mice serum from control MVAgfp vaccinations are protected from RVFV challenge, without logic. It makes all further conclusions from this data infeasible. The one explanation given that Gn-specific humoral responses in response to MVAGn and MVAGnGc would enhance virulence in passively transferred animals is generally interesting but preliminary and should be confirmed by additional experiments.   

Specific comments:

line 111: What is 36% sucrose gradient? Or was it a homogeneous 36% sucrose cushion?

line 211: Judging from V5 Ab staining in Fig 1A the expression level of Gn and Gc is not similar but rather there is significantly more Gn. As a comparison, would be important to show Gn and Gc expression levels from MVAGnGc vector

Fig 1B: The signal from the red channel showing subcellular localization markers is hard to see. The signal should be increased in order to strong conclusions regarding glycoprotein localization to be be assessed.

Fig 2: Please provide numbers of infected animals in the figure legend

Fig. 3B: As presented, the IFA is unconvincing. The fluorescence signal seems unspecific, a distinct negative control is lacking. Would it be possible to do a quantitative measurement of Gn and Gc specific responses such as ELISA? If glycoprotein specific responses exist, why is the immunoprecipitation assay negative?

line 315: Since data for an irrevelant control peptide is not shown, then better not to mention it in the text either.

Figure 5. Is this data from only one animal per group? What is the reason for excluding the pb9 positive control peptide from this experiment? Why do Gc epitopes elicit stronger responses in MVAGc vaccinated as compared MVAGnGc vaccinated splenocytes? I don’t see the value of including the lower range of the standard curve in this figure?

Author Response

RESPONSE TO REVIEWER 2

In the manuscript “MVA vectored vaccines encoding Rift Valley fever virus glycoproteins protect mice against lethal challenge in the absence of neutralizing antibody responses” the authors extend on their previous findings showing protection of mice from RVFV challenge using MVA-vectored RVFV GnGc vaccine. This vaccine induced potent T cell responses against RVFV GnGc while neutralizing antibody responses were minimal. In the current study the authors investigated the individual contribution and nature (T cell or humoral) of Gn or Gc specific responses in mediating protection against RVFV challenge.  The main finding of the study is that immune responses against Gc epitopes are more efficient at eliciting protection against RVFV and similarly to previous work with whole GnGc, are mediated by T cell and not neutralizing antibody responses. However, not all the conclusions of the study are justified by the data presented and significant modifications are required for this study to be publishable.

Major criticisms:

Fig. 4: What is the rationale of choosing the shown Gn and Gc peptides as stimulants. No explanation is given. Similarly, the pb9 positive control peptide is not described. Why are the T cell responses negative for pb9 peptide in MVA-control vaccinations?

These peptides were selected since they were identified and described previously (Lopez Gil et al. 2013 ) as CD8+T-cell epitopes, and worked optimally in ELISPOT assays for spleen cell re-stimulation. The pb9 epitope is described in the Material and Methods sections (see lines 101-102). Also, pb9 peptide has been used by other authors as a positive control for monitoring CTL responses. The negative response of T cells to pb9 stimulation in the control MVA was not unexpected since this virus does not contain the pb9 tag (it was only added to the recombinant RVF sequences).

Fig. 4A: What is the added value of showing PBL responses in the addition to splenocytes in 4B? Since there no replicates and no statistically significant differences can be determined, this data feels redundant.

This was included to check for T-cell responses in the very same mice that were vaccinated and further challenged with the virus (they were not sacrificed to obtain spleens) so we could directly correlate immune responses with protection. Unfortunately we could not provide replicate values due to the limited amount of blood volumes that can be retrieved from a single mouse (the graph therefore represent pools of sera). But we still believe that the data could be informative.

Fig. 6. As presented, the conclusions from experiments shown in Fig.6 are not warranted. The major issue of this experiment is that mice, which have been passively transferred with mice serum from control MVAgfp vaccinations are protected from RVFV challenge, without logic. It makes all further conclusions from this data infeasible. The one explanation given that Gn-specific humoral responses in response to MVAGn and MVAGnGc would enhance virulence in passively transferred animals is generally interesting but preliminary and should be confirmed by additional experiments.  

We agree with this interpretation. It is difficult to understand the nature of the protection provided by the MVA control serum. We were also skeptical but this experiment was repeated and we obtained similar results. Therefore we have a strong argument to think that the immune response to MVA endows the serum with protective capabilities that could reduce the infectivity of RVFV in vivo. We could speculate with high concentrations of complement proteins interacting with the envelope of the infectious particles and inactivating the virus. As the reviewer states this need more experimentation to demonstrate infectivity enhancement but we think this was out of the scope of this work (to show that protection was not related with humoral response).

Specific comments:

line 111: What is 36% sucrose gradient? Or was it a homogeneous 36% sucrose cushion?

The reviewer is right we meant “cushion” instead of “gradient”

line 211: Judging from V5 Ab staining in Fig 1A the expression level of Gn and Gc is not similar but rather there is significantly more Gn. As a comparison, would be important to show Gn and Gc expression levels from MVAGnGc vector

The reviewer is right. It is clear that the levels appear to be slightly lower for Gc. We can include a new panel to show the expression of Gn Gc by rMVAGnGc, as suggested

Fig 1B: The signal from the red channel showing subcellular localization markers is hard to see. The signal should be increased in order to strong conclusions regarding glycoprotein localization to be be assessed.

This can be also corrected in the figures, enhancing the red signal for all the four panels

Fig 2: Please provide numbers of infected animals in the figure legend

This will be included in the figure 2 legend as suggested

Fig. 3B: As presented, the IFA is unconvincing. The fluorescence signal seems unspecific, a distinct negative control is lacking. Would it be possible to do a quantitative measurement of Gn and Gc specific responses such as ELISA? If glycoprotein specific responses exist, why is the immunoprecipitation assay negative?

As the reviewer states the fluorescence observed appears to be unspecific and this probably reinforces out hypothesis of an antibody response being sub neutralizing with very low avidity for native antigens.  This could be the reason why no clear immunoprecipitation of GnGc proteins is seen when using the same serum pools (considering the harsher conditions of the immunoprecipitation buffer). By IFA the serum from the MVA control mice either at 7 or 14 dpi does show a light signal, however the serum from recombinant viruses rMVAGnGc rMVAGn or rMVAGc this signal is clearly stronger. It wold have been great to perform a more quantitative assay but unfortunately we do not have available a quantitative glycoprotein ELISA.

line 315: Since data for an irrevelant control peptide is not shown, then better not to mention it in the text either.

We agree with this suggestion

Figure 5. Is this data from only one animal per group? What is the reason for excluding the pb9 positive control peptide from this experiment? Why do Gc epitopes elicit stronger responses in MVAGc vaccinated as compared MVAGnGc vaccinated splenocytes? I don’t see the value of including the lower range of the standard curve in this figure?

In this experiments the re-stimulated spleen cells were pooled from four mice used in the  spleen ELISPOT assays. The pb9 peptide was not included since the three glycoprotein peptides worked well and were considered positive controls as seen previously. We do not know why Gc epitopes display stronger responses that GnGc, but this fact appears to correlate well with the stronger cell responses observed in the other assays.

Round 2

Reviewer 2 Report

The problem with the passive transfer experiment persists and unless further control experiments are performed, authors should omit Fig. 6 from the manuscript. The authors admit that passively transferred rMVAgfp negative control serum seems to protect from RVFV challenge, but do not provide an adequate explanation for this phenomenon. Would serum obtained from any naive BALB/c mice have the same effect? In addition, would be important to show positive control (RVFV challenge without passively transferred serum) in the same experiment.

Author Response

Response to reviewer 2:
"The problem with the passive transfer experiment persists and unless further control experiments are performed, authors should omit Fig. 6 from the manuscript. The authors admit that passively transferred rMVAgfp negative control serum seems to protect from RVFV challenge, but do not provide an adequate explanation for this phenomenon. Would serum obtained from any naive BALB/c mice have the same effect? In addition, would be important to show positive control (RVFV challenge without passively transferred serum) in the same experiment."

Since we think that the passive transfer experiments provide support to our main findings (i.e: the capacity of MVA vectored vaccines to provide protection when no neutralizing antibodies are detected) we would like to maintain Fig 6 in this work. As the reviewer suggest, to avoid omit this figure we have included further control experiments. In this case we have included an additional control group (mice passively transferred with serum from HuAd5 immunized mice taken at 14 dpi). This can be considered an irrelevant serum since the adenovirus vector does not carry neither specific RVFV antigens nor MVA antigens. To include this novel control we have also replicated the experiment of virus titration to detect the presence of infectious virus in blood which is now shown in the new figure 6. Additionally we have included as supplementary data the weight change variation in the inoculated mice to provide an estimation of the clinical display upon infection. We think this data may now show that the mice from the rMVAgfp negative control became infected although the outcome of this infection was not as lethal. To provide an explanation from this phenomenon would need additional experiments that we cannot afford in the short term.